# Spatio-temporal Heterogeneous Federated Learning for Time Series Classification with Multi-view Orthogonal Training

### Chenrui Wu
College of Computer Science
Zhejiang University
Hangzhou, China
chenruiwu@zju.edu.cn

### Haishuai Wang*
College of Computer Science
Zhejiang University
Hangzhou, China
haishuai.wang@zju.edu.cn

### Xiang Zhang
Department of Computer Science
The University of North Carolina at
Charlotte, USA
xiang.zhang@charlotte.edu

### Zhen Fang
Australian AI Institute
University of Technology Sydney
Sydney, Australia
zhen.fang@uts.edu.au

### Jiajun Bu
College of Computer Science
Zhejiang University
Hangzhou, China
bjj@zju.edu.cn

## Abstract

Federated learning (FL) is undergoing significant traction due to its ability to perform privacy-preserving training on decentralized data. In this work, we focus on sensitive time series data collected by distributed sensors in real-world applications. However, time series data introduce the challenge of dual spatial-temporal feature skew due to their dynamic changes across domains and time, differing from computer vision. This key challenge includes inter-client spatial feature skew caused by heterogeneous sensor collection and intra-client temporal feature skew caused by dynamics in time series distribution. We follow the framework of Personalized Federated Learning (pFL) to handle dual feature drifts to enhance the capabilities of customized local models. Therefore, in this paper, we propose a method FedST to solve key challenges through orthogonal feature decoupling and regularization in both training and testing stages. During training, we collaborate time view and frequency view of time series data to enrich the mutual information and adopt orthogonal projection to disentangle and align the shared and personalized features between views, and between clients. During testing, we apply prototype-based predictions and model-based predictions to achieve model consistency based on shared features. Extensive experiments on multiple real-world classification datasets and multimodal time series datasets show our method consistently outperforms state-of-the-art baselines with clear advantages.

## CCS Concepts

• **Computing methodologies → Distributed artificial intelligence**; • **Mathematics of computing → Time series analysis**.

*Corresponding author: Haishuai Wang.

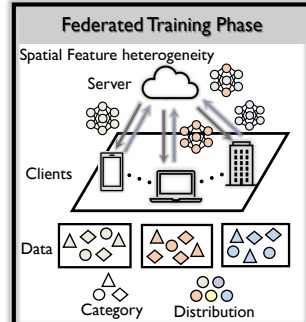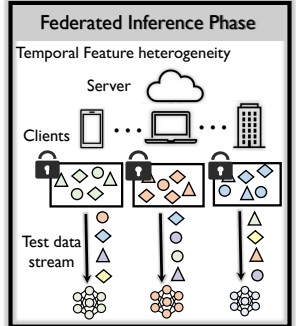

**Figure 1: Illustration of the spatial-temporal heterogeneous federated learning setting. Data feature skew exists among clients during the training stage. For test time, unlabeled test data also shows a domain gap due to non-stationarity.**

## Keywords

Federated Learning, Time Series, Feature Distribution Shift

**ACM Reference Format:**
Chenrui Wu, Haishuai Wang, Xiang Zhang, Zhen Fang, and Jiajun Bu. 2024. Spatio-temporal Heterogeneous Federated Learning for Time Series Classification with Multi-view Orthogonal Training. In *Proceedings of the 32nd ACM International Conference on Multimedia (MM '24), October 28– November 1, 2024, Melbourne, VIC, Australia.* ACM, New York, NY, USA, 10 pages. https://doi.org/10.1145/3664647.3680733

## 1 Introduction

Nowadays, machine learning (ML) is undergoing a paradigm shift from cloud data centers to distributed edges [19, 29]. With the development of mobile Internet of Things (IoT) [61] and multimedia computing platforms [16, 24, 48, 72], a large amount of valuable time series data is recorded by distributed smart devices or entities. Time series data refers to a series of data points or observations arranged in chronological order. It has been widely used in anomaly detection [23, 26], activity recognition [13, 74], weather forecasting [7], healthcare diagnosis [59, 66], recommendation system [21, 22, 68], and emotion analysis [30]. One of the major research topics is time series classification (TSC) by deep neural networks (DNN) [41, 58].

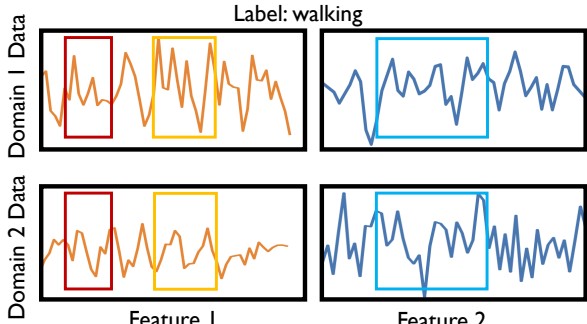

**Figure 2: Visualization of feature distribution shifts of time series data by two samples with the same label space but of different domains. The two samples hold the same label space but show domain-variant representation on both dimensions of features (monitored multi-dimension time series). The data is based on WIDSM [28]**

However, due to the sensitivity and privacy issues of users' data, the model training in a centralized data way faces challenges in practice. Traditional centralized machine learning methods require all data to be concentrated on a central server for training, which not only increases the overhead of data transmission but also raises potential risks of privacy leaks.

To tackle the challenges of data confidentiality and communication efficiency, federated learning (FL) [1, 25, 32, 44, 60, 62, 71] emerged as a promising distributed training paradigm. Federated learning can collaboratively train from massive multi-source data without exchanging the original data, retaining data ownership [34] while reducing the communication burden [43]. Specifically, edge devices preserve their private data locally, and federated learning mainly realizes training a robust model by aggregation and distribution of local models through multiple rounds of communication.

Although FL shows promising prospects in data collaboration, it still faces the challenge of data heterogeneity in practice [44]. Time series data collected by sensors often come from different individuals and institutions, where the client's local data is non-independent and equally distributed (non-IID). Non-IID data can lead to model drifts and catastrophic forgetting of global knowledge in federated learning, which further leads to model performance degradation and slow convergence. Data heterogeneity in federated learning can be categorized into label space heterogeneity and feature space heterogeneity. Much work has focused on the problem of non-IID in the label space heterogeneity among clients [1, 6, 31, 32, 35]. Compared with the more obvious label distribution deviation caused by class imbalance, the feature skew data in federated learning is more hidden and more difficult to overcome [5, 33, 77, 78].

Although general feature distribution skew has been explored in FL, it becomes more challenging when applied to time series data beyond image processing [5, 33, 77, 78], i.e., the data exhibit dual feature heterogeneity in both spatial and temporal perspectives.

**Challenge 1: Spatial feature heterogeneity.** In real-world deployment, the time series data collection methods of different devices and individuals vary greatly, resulting in cross-client feature skew [10]. As shown in Figure 2, two samples share the label space but exhibit heterogeneity in feature space. In the FL context, each client may become a domain, and the complex dependence of

time steps makes it difficult to extract invariant features [3], hindering the cross-domain generalization ability of FL models. From another perspective, time series have two perspectives: time view and frequency view. Domain shift may occur in both time view and frequency view. It is also possible to shift the time feature while leaving the frequency feature relatively unchanged, as shown in Figure 3. Explicitly modelling time and frequency domain features becomes more challenging.

**Challenge 2: Temporal feature heterogeneity.** The non-stationarity of time series data has been an under-addressed challenge [11, 27, 40, 76]. Since the time series data is observation data discretely sampled by sensors at a certain recording frequency, its future data (horizons) may show different feature distributions from the past observations (look-backs) [11, 27]. In the FL, it reflects in the distribution shifts between training sample features and testing sample features [18, 45, 52, 57]. The feature drifts of test samples make the original model unable to perform ideally on the changing dynamics. Accessing the original data will cause huge computational costs for complete retraining of the model. How to realize test time adaption for robust FL deployment is under-explored [20, 54].

Encountering the dual challenges of spatial-temporal heterogeneous federated learning, we need to introduce a training scheme: personalized federated learning (pFL) [6, 8, 9, 31, 32, 55, 63, 64] which is tightly associated with dual feature-skewed FL and test time adaption. Differing from vanilla generalized FL, pFL aims to enhance the capability of clients' personalized models during the local test phase. From the perspective of spatial feature heterogeneity, a well-collaborated generic model finally needs to fit each local test set's domain. Thus, previous works about feature-skewed FL were often presented as a uniform challenge along with pFL [5, 33, 42, 54, 56, 78]. Besides, some pFL works mainly focus on label distribution shifts [6, 8, 31, 32, 53]. From the point of view of temporal feature heterogeneity, clients' time series data shows concept drifts during test time. The optimization goal of pFL is that the trained model performs better on each client's test samples, which coincides with the test-time adaption objective of fine-tuning the model online to improve test-time performance.

To overcome these challenges, in this paper, we propose FedST, a two-stage updated federated learning framework to solve spatial-temporal feature heterogeneity. The framework mainly consists of two modules: orthogonal training and consistency testing. In the training phase, we propose an orthogonal decoupling of cross-view and cross-client representations to extract and utilize shared and private features. In the testing phase, we align the predictions of the prototype and the predictions of the model based on the orthogonal subspace, minimizing uncertainty and fine-tuning representations.

Specifically, for the feature-skewed FL, previous work has discovered the importance of decoupling personalized knowledge and global knowledge. Multi-model methods have been used to disentangle different features in images [12, 42, 70] for promoting local personalization. For the distributed feature shifts of time series, we focus on not only the decomposition of cross-domain features between clients but also the decomposition of view-wise features of time series. Time series data can be converted into multimodal information, and supplementary information from different views can be used to effectively analyze time series data [73, 75]. Multi-view learning can extract extra information [38, 73, 75] of time

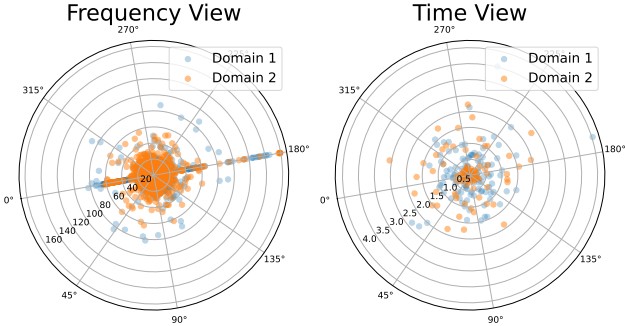

**Figure 3: Visualization of feature distribution about time view and frequency view. The displayed data are sensor records from two walking persons (domains) in a public dataset. Corresponding polar coordinates of frequency features show that frequency features are more domain-invariant than time features in some cases. The data is from WISDM [28].**

series than single-view in the latent space, as exhibited in Figure 3. We hope that the optimization objective is to improve the orthogonal between shared features and private features of the same view, the private features between different views, and private features between clients. Therefore, we can capture the private features and shared features between views, and the personalized knowledge and global knowledge between clients through the orthogonal space of factorization. We adopt contrastive learning and prototype-based methods to achieve our cross-client and cross-modal orthogonal training. In the federated inference phase, the visited model learned a feature representation ability that fits the local domain with global knowledge. However, since the feature concept of the time series drifts over time, the model may not be able to generate accurate representations for newly exposed samples. We leverage aligning the predictions of the prototype and the predictions of the model based on orthogonal subspaces to maintain model consistency and representation quality. We summarize our contributions as follows:

- We consider a unique spatial-temporal feature skew challenge tailored for time series in the real-world application of federated learning.
- We propose FedST which leverages orthogonal latent space to disentangle client-level and view-level features, and adopts prototype-based consistency for test time adaption.
- Extensive experimental results, FedST has achieved superior empirical performance better than SOTA solutions for pFL, feature-skew FL and FL for time-series.

## 2 Related Work

### 2.1 Federated Learning for Time Series

Recently, many deep learning methods have been adopted to process distributed sensitive time series data monitored by private sensors. Federated learning, as a significant decentralized deep learning paradigm, has been widely used to analyze these time series data. In [39] an attention mechanism-based convolutional neural network-long short-term memory (AMCNN-LSTM) model is proposed for industrial anomaly detection. FedTSC [36] presents a two-level secure mechanism for time series classification with

secure feature extraction and secure model training. They emphasize security and interpretability. EFDLS [65] focuses on a multi-task time series classification setting and proposes a knowledge distillation-based framework. MetePFL [7] proposes a prompt learning method for weather forecasting with foundation models. However, it holds a strong hypothesis that each client can fine-tune a foundation model, which is not universally practical in mobile computing scenarios for time series. FLAMES2Graph [69] also targets the interpretability of time series classification. They adopt a Multivariate Highly Activated Period (MHAP) evolution graph aggregation for better interpretable performance. It is notable that these works mostly focus on the general performance of time series classification or forecasting, ignoring the challenges of the time series data modality itself, such as the time and frequency feature and non-stationarity that this paper focuses on. Also, they did not consider the challenge resulting from the implementation of federated learning but adopted time series as a general downstream task.

### 2.2 Feature-Skewed Federated Learning

Feature-skewed federated learning focuses on improving the customized capability of clients' local models under diverse feature spaces or domains held by clients. As pFL methods for label skew, model decoupling [6, 20] and prototypical methods [55, 56] are also widely adopted for feature-skewed FL. FedBN [33] preserves the batch normalization layers, aggregating other parameters to overcome feature concept shifts among clients. AlignFed [78] performs a reverse aggregation strategy of decoupled personalized federated learning for label skew. They craft personalized feature extractors and generalized classifiers. FedPCL [56] adds contrastive loss as an incremental approach of FedProto [55]. The pre-trained parameters are also considered to improve the initial performance of FL. DFL [42] aims to disentangle shared and private features by multi-branch feature extractor and applies mutual information loss to optimize the training. Differing from model decoupling, FedFA [77] and FedRDN [67] focus on the data itself, applying data augmentation to enrich data representations. With expanded feature representation, models are more robust to different underlying distributions across clients. However, previous works mainly emphasize the distribution shifts in the computer vision context. The spatial feature heterogeneity of time series data is under-explored.

### 2.3 Test-Time Adaptation

Test-time adaptation, which has been widely investigated in the fields of computer vision [17, 45, 52, 57], aims to improve model performance on unknown target domain data during the test time. FedTHE [20] as a pioneer work, explore the ID and out-of-distribution (OOD) situations that clients will encounter during federated deployment. They decouple the original model into a global head and personalized head and perform robust cross-entropy. They also enforce feature space alignment via prototypes. FedICON [54] adopts contrastive learning to FedProto [55], making the predictions of different augments of a sample more stable to achieve robust updates during test time. These works concentrate on the domain shifts and common corruptions of image processing, but we extend to the unique distribution drifts of time series.

Feature distributions of time series in the real world are often dynamic and change over time, especially for scenarios such as weather forecasting and clinical detection. Many works have focused on it. Non-stationary [40] proposed a new transformer that contains Series Stationarization and De-stationary Attention modules. Dish-TS [11] introduces a dual-conet framework to learn the distribution of input and output spaces separately, naturally capturing the distribution differences between the two spaces. OneNet [76] uses reinforcement learning to train two networks to balance temporal correlation and cross-variable dependence. RevIN [27] normalizes the input time series to fix its distribution in terms of mean and variance, then returns the output to the original distribution. [26] first propose test-time adaption in unsupervised time series anomaly detection. They utilize trend estimates and normal instances based on the model prediction itself for model updates.

## 3 Preliminaries

### 3.1 Problem formulation

**Basic setting.** In this work, we introduce a pFL problem with spatial-temporal heterogeneous feature-skewed data $\mathcal{D}$. In the FL system, we consider each $k \in \{1, 2, \cdots, K\}$ client holding a private time series dataset $\mathcal{D}_k = \{(x_{i,k}, y_{i,k})\}_{i=1}^{n_k}$, where $n_k$ denotes the number of samples of local dataset as $n_k = |\mathcal{D}_k|$. For a given time series sample $x_i \in \mathbb{R}^{T \times d}$ includes $d$-demesions variate over $T$ time points, and it's corresponding label $y_i \in C = \{1, 2, \cdots, C\}$. Given the local time series datasets for $i$–th client, $\mathcal{D}_i$ can be divided into training set $\mathcal{D}_i^{train}$ and test set $\mathcal{D}_i^{test}$.

**Spatial heterogeneity.** For arbitrary client $i$ and client $j$, the feature skew heterogeneity across clients can be denoted as:

$$P_i(x|y) \neq P_j(x|y), \; where \; C_i = C_j, \; \forall i \neq j, \; i, j \in [K], \quad (1)$$

where $P_i(x|y)$ refers to the conditional probability of $x$ given $y$ for client $i$. The label space of $C_i$ remains homogeneous, and $P_i(y)$ is the same. Otherwise, it leads to a label distribution shift $P_i(y) \neq P_j(y)$.

**Temporal heterogeneity.** Consider the temporal context of time series data, the training sample $\{x_{t-L:t}\} = [x_{t-L+1}, \cdots, x_t] \in \mathcal{D}_i^{train}$ follow a probability distribution $\mathbb{P}_i^{train}$ in the training phase. $L$ refers to the recorded time steps of time series data. The test set $\{x_{t:t+H}\} = [x_{t+1}, \cdots, x_{t+H}]$ also follow the distribution $\mathbb{P}_i^{test}$. The underlying distribution $\mathbb{P}_i$ is non-stationary, where $\mathbb{P}_i^{train} \neq \mathbb{P}_i^{test}$ can be formulated as:

$$|d(\mathbb{P}_i^{train}), d(\mathbb{P}_i^{test})| \geq \delta, \quad (2)$$

where $\delta$ is the threshold and $d(\cdot)$ is the distance metric to evaluate the distributions' divergence, such as KL divergence.

**Optimization objective.** We denote $\theta$ as the basic model architecture with two-part: feature extractor $f$ parameterized by $\theta_f$ and prediction head $c$ parameterized by $\theta_{cls}$. The typical FL considers a generalized optimization problem, to minimize the empirical risks:

$$\min \mathcal{L}(\theta) = \sum_{i=1}^{K} \frac{|\mathcal{D}_i|}{|\mathcal{D}|} \mathcal{L}_i(\theta), \forall i \in [K]. \quad (3)$$

In contrast to traditional generalized FL, pFL aims to improve every client's personalized models. Clients pursue adapting to each local data distribution collaboratively. The client-wise personalized learning objective can be formulated as:

$$\min_{\{\theta_1, \theta_2, \cdots, \theta_K\} \in \Theta} \mathcal{L}(\Theta) = \sum_{i=1}^{K} \frac{|\mathcal{D}_i|}{|\mathcal{D}|} \mathcal{L}_i(\theta_i), \forall i \in [K], \quad (4)$$

where clients' personalized objectives can be computed by loss function and reach a better performance to tackle the cross-client feature skew and test time feature skew.

### 3.2 Time series frequency features

The Discrete Fourier Transform (DFT) is often applied to transform signals in the time domain into the frequency domain. With different domains, the cognitive perspective of the same thing also changes. From a new view of the frequency domain, more representations can be revealed. For each dimension of a time series sample $x$, the DFT process can be denoted as:

$$X[k] = \sum_{t=0}^{T-1} x[t] \cdot e^{-i \frac{2\pi}{T} kt}, \quad (5)$$

where $X[k]$ is the $k$-th frequency component in the frequency domain, $x[t]$ is the $t$-th data point in the time series, $T$ is the time step (total number of data points), $k$ is the current frequency index, $i$ is the imaginary unit.

## 4 Method

### 4.1 Overview

In this section, we describe the proposed novel federated learning method: FedST. We consider two main processes: federated training and federated inference. In the training phase, to tackle the feature distribution shifts across clients, we adopt an orthogonal factorization way. By projecting features to orthogonal subspace, we align the shared feature between time and frequency view, and local and global representations. The communication between clients and the server includes spatial model aggregation and global prototype aggregation. Considering the feature distribution shift in the test time of time series data, we craft an adaption objective to fine-tune the representation and make prediction more confident. Figure 4 visualizes the training procedure of both phases.

### 4.2 Client Update

**Time-Frequency Feature Encoder.** To build multi-view collaborative training, two single-view encoders are first developed to map the input time series data into two latent spaces. With recent advances in time series learning, many network structures have been proposed to extract time series features. The encoder network structure is not the original contribution of this paper, we focus on handling the skew problem of extracted features. Given a sample $x_i$, we adopt a time encoder $\theta_f^T$ and a frequency encoder $\theta_f^F$ to obtain the features $z_i^T$ and $z_i^F$, respectively. We conduct some warm-up rounds with the classifier $\theta_{cls}$, and aggregate the initial global prototypes, defined as:

$$\boldsymbol{p}_{k,c}^T = \frac{1}{|\mathcal{D}_{k,c}|} \sum_{(x_{i,k}, y_{i,k}) \in \mathcal{D}_{k,c}} f(\theta_f^T; x_{i,k}), \; \forall c \in C, \quad (6)$$

where $\boldsymbol{p}_{k,c}^T$ refers to the mean of time domain representations for a class $c$. With the fusion of participated clients, the global prototypes

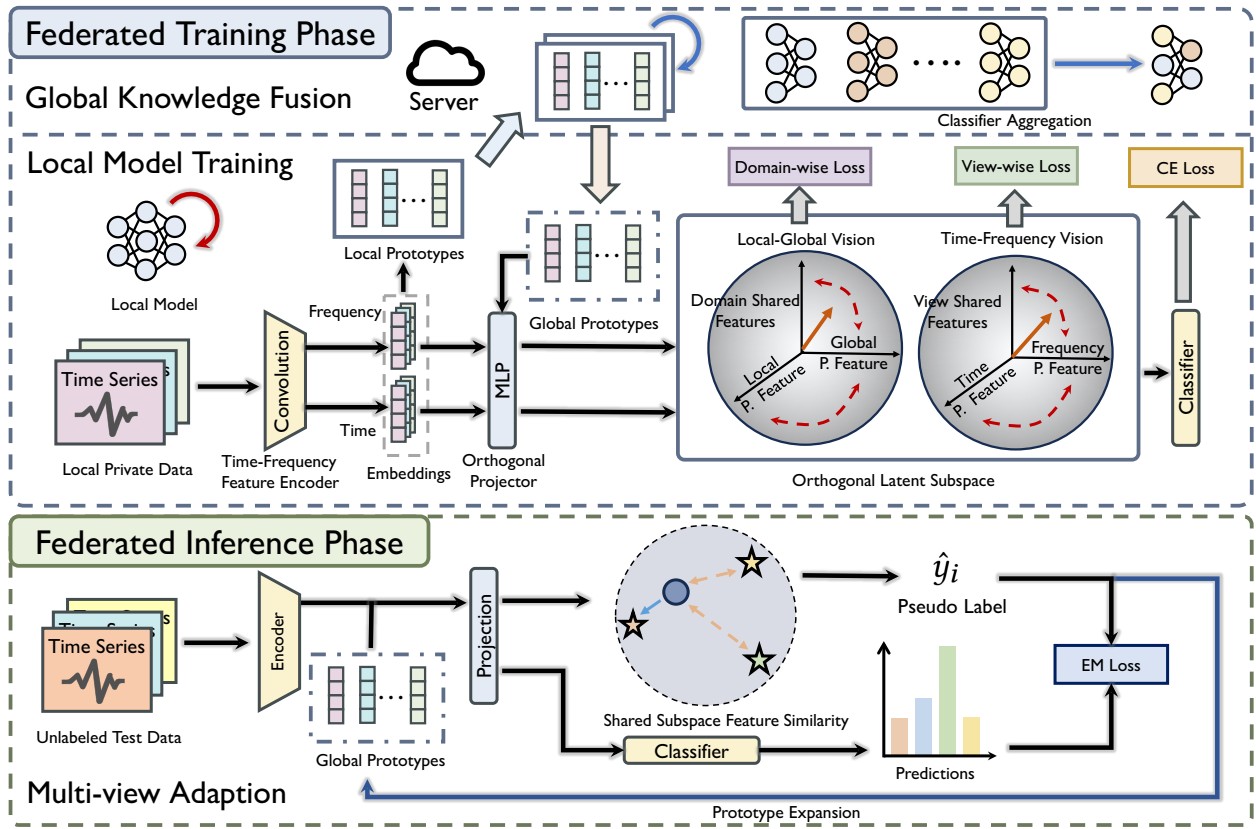

**Figure 4: The framework overview of the proposed FedST.**

as a global knowledge, written as:

$$\boldsymbol{p}_c^T = \sum_{k=1}^{K} \frac{|\mathcal{D}_k|}{|\mathcal{D}|} \boldsymbol{p}_{k,c}^T, \forall k \in [K], \qquad (7)$$

where the set of global prototypes can be obtained as $\{\boldsymbol{p}_c^T\}_{c \in C}$, and frequency prototype $\{\boldsymbol{p}_c^F\}_{c \in C}$ can also be derived.

**Orthogonal Space Factorization.** We first recall the definition of orthogonality, given two vectors $v$ and $u$:

$$v \cdot u = v^\top u = u^\top v = 0, \qquad (8)$$

where the product of two vectors is zero, then we derive they are orthogonal. Given a vector space $\mathcal{V}$, we can define two orthogonal subspaces $\mathcal{V}^s$ and $\mathcal{V}^p$. Thus, any vector in space can be disentangled into two orthogonal subspaces via projection matrixes $P_i$. Inspired by previous literature [4, 12, 38], for shared subspace and personalized subspace, we ensure the orthogonality of the projection matrix: $P_s^\top P_s = I$, $P_p^\top P_p = I, P_s^\top P_p = 0$, $P_p^\top P_s = 0$. Given a intermediate layer $l$ of feature extractor $f_l$ with $L$ hidden layers, the optimization objective of gradients can be decomposed as:

$$g_l^{\{s,p\}} = \frac{\partial l}{\partial f_l} = \left(\frac{\partial l}{\partial f_L}\right)\frac{\partial f_L}{\partial f_l} = \left(\frac{\partial \phi_{P_{\{s,p\}}}}{\partial f_L}\frac{\partial l}{\partial \phi_{P_{\{s,p\}}}}\right)\frac{\partial f_L}{\partial f_l}$$

$$= \left(P_{\{s,p\}}\frac{\partial l}{\partial \phi_{P_{\{s,p\}}}}\right)\prod_{k=l}^{L-1}\frac{\partial h_{k+1}}{\partial h_k} = g_L^{P_{\{s,p\}}}\prod_{k=l}^{L-1}D_{k+1}\theta_f^{k+1}, \qquad (9)$$

where $D_k$ denotes a diagonal matrix that embodies the Jacobian matrix associated with the pointwise nonlinearity. $P_i$ refers to the

projection matrix, $\phi_P$ denotes the product of $P_i$ and $l$-layer parameters. Due to the orthogonal gradient updates, we can project features into orthogonal subspaces and disentangle them. After receiving the output of time-frequency encode, each view embedding $\boldsymbol{h}_i^T, \boldsymbol{h}_i^F$ are projected by a non-linear multilayer perceptron (MLP) projector, into two orthogonal embeddings: personalized feature embeddings $\boldsymbol{h}_i^{T,p}, \boldsymbol{h}_i^{F,p}$ and shared feature embeddings $\boldsymbol{h}_i^{T,s}, \boldsymbol{h}_i^{F,s}$.

**View-wise Orthogonality Loss.** Inspired by [73, 75], we convey wealth to mining the mutual information among different time series views and enhance representation training by pulling in shared features and pushing away personalized features. In the orthogonal latent subspace, we adopt a contrastive way to achieve this objective, written as:

$$\mathcal{L}_{vol}^T = -\frac{1}{|\mathcal{D}_k|}\sum_{x_i \in \mathcal{D}_k}\left(\underbrace{\log\frac{\exp(|\boldsymbol{h}_i^{T,s} \cdot \boldsymbol{h}_i^{F,s}|/\tau)}{\sum_{x_j \in \mathcal{B}}\exp(|\boldsymbol{h}_i^{T,s} \cdot \boldsymbol{h}_j^{F,s}|/\tau)}}_{\text{shared features align}} - \right.$$

$$\left.\underbrace{\log\frac{\exp(|\boldsymbol{h}_i^{T,p} \cdot \boldsymbol{h}_i^{F,p}|/\tau)}{\sum_{x_j \in \mathcal{B}}\exp(|\boldsymbol{h}_i^{T,p} \cdot \boldsymbol{h}_j^{F,p}|/\tau)}}_{\text{personalized features disentangle}}\right), \qquad (10)$$

where $\mathcal{B}$ denotes a batch in the learning process, $\tau$ refers to a temperature parameter to control the tolerance. The corresponding contrastive loss for frequency view $\mathcal{L}_{vol}^F$ can be obtained in the same way. Within a single view, we also enforce the projection

in orthogonal subspace to minimize the angular between shared feature and personalized feature. We define the private orthogonal loss for an unimodal as:

$$\mathcal{L}_{pol}^T = \frac{1}{|\mathcal{D}_k|} \sum_{x_i \in \mathcal{D}_k} |h_i^{T,s} \cdot h_i^{T,p}|, \tag{11}$$

where we are able to formulate frequency view loss $\mathcal{L}_{pol}^F$ as well. In this way, orthogonality between these two features can be strengthened. The local encoder and projector possess the capacity to acquire invariances and variances of shared and personalized features.

**Domain-wise Orthogonality Loss.** As discovered in [12, 54, 70], the domain gap between clients is vital for feature-skewed FL. Learning to decouple the global feature and personalized feature remains to be investigated. In this work, we continue to adopt orthogonal subspace projection for global knowledge and local embedding. We utilize global prototypes as global knowledge and project them into orthogonal subspace as $\widetilde{P}_c^{T,s}$ and $\widetilde{P}_c^{T,p}$, as well as frequency view $\widetilde{P}_c^{F,s}, \widetilde{P}_c^{F,p}$. We facilitate each sample to be invariant in the shared feature space, and orthogonal in the personalized feature space. We perform cross-client orthogonality constraint with the loss function as below:

$$\mathcal{L}_{dol}^T = -\frac{1}{|\mathcal{D}_k|} \sum_{x_i \in \mathcal{D}_k} |h_i^{T,s} \cdot \widetilde{p}_{y_i}^{T,s}| + \frac{1}{|\mathcal{D}_k|} \sum_{x_i \in \mathcal{D}_k} |h_c^{T,p} \cdot \widetilde{p}_{y_i}^{T,p}|, \tag{12}$$

We summarize the total orthogonal loss for each view $\{T, F\}$ as:

$$\mathcal{L}_{opl}^{\{T,F\}} = \lambda_1 \mathcal{L}_{dol}^{\{T,F\}} + \lambda_2 (\mathcal{L}_{vol}^{\{T,F\}} + \mathcal{L}_{pol}^{\{T,F\}}). \tag{13}$$

The general optimization objective cross-entropy loss to minimise empirical loss is formulated as follows:

$$\mathcal{L}_{ce} = -\frac{1}{|\mathcal{D}_k|} \sum_{i=1}^{|\mathcal{D}_k|} y_i \log(f(\boldsymbol{\theta}_{cls}; z_i)). \tag{14}$$

Finally, the overall loss function is denoted as:

$$\mathcal{L} = \mathcal{L}_{ce} + \lambda(\mathcal{L}_{opl}^T + \mathcal{L}_{opl}^F), \tag{15}$$

## 4.3 Server Aggregation

Vanilla Fedavg [44] aggregates clients' models based on local dataset size as $\boldsymbol{\theta}^{(t+1)} = \sum_{k \in [K]} \frac{|\mathcal{D}_k|}{|\mathcal{D}|} \boldsymbol{\theta}_k^{(t)}$. Averaged aggregation strategy fits the goal of generalization, not for personalization. For label-skewed pFL, model decoupling has been widely adopted, where the classifier is regarded as the source of model shifts. So, a shared feature extractor with a personalized classifier stands out as a popular structure [6, 8, 20, 70]. In contrast to label skew, a reversed aggregation is commonly implemented with a personalized feature extractor with a shared classifier [12, 42, 54, 78] for feature skew. In this work, we aggregate the classifier $\boldsymbol{\theta}_{cls}$ and preserve the time-frequency encoder $\boldsymbol{\theta}_f^{\{T,F\}}$ and projector to fit the feature-skewed personalization task, as previous work $\boldsymbol{\theta}_{cls}^{(t+1)} = \sum_{k \in [K]} \frac{|\mathcal{D}_k|}{|\mathcal{D}|} \boldsymbol{\theta}_{cls,k}^{(t)}$. In addition, clients also upload local time-frequency prototypes $\{\boldsymbol{p}_c^{\{T,F\}}\}_c^C$ for global knowledge fusion [55, 56, 78], denoted as:

$$\boldsymbol{p}_c^{(t+1)} = \sum_{k \in [K]} \frac{|\mathcal{D}_k|}{|\mathcal{D}|} \boldsymbol{p}_{k,c}^{(t)}, \forall c \in C. \tag{16}$$

**Table 1: Summary of datasets used in the experiments.**

|  | HAR | HHAR | WISDM | Sleep-EDF | Epilepsy |
|---|---|---|---|---|---|
| # Train | 7352 | 12716 | 1350 | 35503 | 9200 |
| # Test | 2947 | 5218 | 720 | 6805 | 2300 |
| Length | 128 | 128 | 128 | 3000 | 178 |
| # Subjects | 30 | 9 | 30 | 20 | 500 |
| Channel | 9 | 3 | 3 | 1 | 1 |
| # Class | 6 | 6 | 6 | 5 | 2 |

## 4.4 Client Inference

In the federated inference phase, training data is no longer approachable. The feature distributions of coming test samples show dynamic shifts due to the non-stationary time series. In this case, gradually updating the trained model with unsupervised data can help the model adapt to the changing feature concept.

**Prototypical Orthogonal Adaption.** Given an unlabeled test sample $x_i$ during adaption, we can obtain the projected feature $h_i^s, h_i^p$ as a shared feature and personal feature for both time and frequency view. We maintain the pre-trained global prototypes $\{\boldsymbol{p}_c\}_{c \in C}$ as a prototypical classifier. Instead of computing cosine or Euclidean similarity to get the nearest prototype as a pseudo label, we produce the pseudo label via a minimized angler of the shared feature in the orthogonal subspace:

$$\hat{y}_i = \arg \min_c \sigma(\{|h_i^s \cdot \widetilde{p}_c^s|\}_{c=1}^C), \tag{17}$$

where $\sigma(\cdot)$ is the normalized prediction by a softmax function. We align the predictions of the prototype (in the orthogonal subspace) with the predictions of the model to ensure that the extracted feature distribution of the current test data should be consistent with the feature distribution of all previous data of the same class. Thus, the loss to maintain consistency and align representation is defined as:

$$\mathcal{L}_{poa}(p_i, \hat{y}_i) = -\sigma(p_i) \log \hat{y}_i, \tag{18}$$

where we omit the notation for time and frequency views, the total loss is calculated by the sum of $\mathcal{L}_{poa}^T$ and $\mathcal{L}_{poa}^F$. The process mentioned above is undergoing for both views. By minimizing the entropy, the model is able to generate more accurate representations of samples in the target domain. We then update the prototypes with new data, gradually fitting the dynamic distributions.

## 5 Experiments

## 5.1 Experimental Setup

**Datasets.** We use five real-world time-series datasets for evaluation, which are commonly used in domain adaption and other feature drifts works for time series [15, 37, 47], involving human activity recognition **HAR** [13], **HHAR** [51], **WISDM** [28] and Electroencephalogram (EEG) signal data: **Sleep-EDF** [14], **Epilepsy** [2]. The summary is given in Table 1.

**Baselines.** We take four groups of methods as baselines. (1) typical FL and pFL for label distribution shifts: FedAvg [44], Fed-Prox [32], FedDyn [1], FedRoD [6], FedProto [55] (2) pFL for feature distribution shifts: FedBN [33], AlignFed [78], FedFA [77] (3) pFL for test time adaption: FedTHE [20], FedICON [54] (4) FL tailored for time series: FLAMES2Graph [69].

**Data distributions.** We consider dual feature distribution shifts.

- **Spatial feature heterogeneity**: For spatial feature heterogeneity, we adopt the general feature-skewed setting in pFL. We treat each entity in time series data as a domain, the

**Table 2: Results of natural temporal feature skew in terms of personalization accuracy (%) of local models on five datasets under different numbers of clients ($K$) (domains). Green/bold fonts highlight the best baseline/our method.**

| Dataset | HAR | | HHAR | | WISDM | | Sleep-EDF | | Epilepsy | |
|---|---|---|---|---|---|---|---|---|---|---|
| Number of Clients ($K$) | 30 | 60 | 9 | 18 | 30 | 60 | 20 | 40 | 100 | 200 |
| FedAvg[2017] | 81.29±0.64 | 76.15±1.41 | 63.07±0.28 | 60.45±2.21 | 64.83±1.76 | 59.51±1.48 | 76.02±0.56 | 65.70±1.35 | 85.52±2.46 | 72.15±3.57 |
| FedProx[2020] | 83.04±0.23 | 81.46±1.26 | 66.94±0.74 | 62.88±1.42 | 63.34±0.77 | 61.09±0.81 | 74.53±1.78 | 66.02±1.49 | 87.15±1.58 | 71.90±1.56 |
| FedDyn[2021] | 81.98±0.39 | 72.40±2.46 | 64.33±0.77 | 62.75±1.80 | 58.47±1.58 | 62.50±0.61 | 76.86±1.06 | 62.88±2.27 | 84.56±2.96 | 73.45±2.49 |
| FedProto[2022] | 82.16±0.37 | 78.41±0.84 | 66.20±0.42 | 64.49±0.18 | 69.97±1.94 | 66.45±1.52 | 77.36±0.23 | 59.56±1.07 | 85.82±1.44 | 73.87±3.27 |
| FedRoD[2022] | 83.53±0.70 | 82.78±1.13 | 67.42±1.28 | 66.72±0.12 | 66.24±0.62 | 65.91±1.39 | 75.36±0.64 | 65.23±1.65 | 88.41±0.83 | 75.20±1.59 |
| FedBN[2021] | 86.21±0.31 | 82.55±0.44 | 69.04±0.75 | 65.39±0.92 | 62.79±1.06 | 63.90±2.03 | 77.58±1.61 | 70.93±2.54 | 86.63±0.29 | 74.06±1.37 |
| AlignFed[2022] | 85.98±0.66 | 84.51±0.20 | 69.95±0.37 | 68.26±0.53 | 67.01±1.26 | 70.84±0.49 | 76.57±1.86 | 72.98±2.04 | 90.06±1.33 | 76.12±1.65 |
| FedFA[2023] | 87.90±0.29 | 86.09±0.88 | 72.51±1.66 | 68.82±1.97 | 68.26±0.55 | 72.59±1.04 | 79.56±0.86 | 70.24±1.09 | 92.04±1.87 | 86.52±2.57 |
| FedTHE[2023] | 86.76±1.15 | 79.61±0.98 | 70.45±0.25 | 62.01±2.25 | 66.17±1.28 | 71.04±1.09 | 77.22±1.30 | 68.09±0.45 | 89.36±1.75 | 76.41±2.59 |
| FedICON[2023] | 88.15±0.87 | 84.50±1.73 | 71.37±1.04 | 67.93±1.32 | 70.83±1.45 | 70.52±2.61 | 79.97±1.44 | 72.41±2.59 | 92.45±1.33 | 85.37±1.70 |
| FLAMES2Graph[2023] | 87.72±1.06 | 85.69±1.35 | 74.24±1.16 | 70.52±2.35 | 69.92±1.70 | 63.99±1.04 | 81.57±1.21 | 71.44±1.29 | 91.30±1.76 | 80.54±1.89 |
| **Our FedST** | **93.02±0.65** | **90.53±0.45** | **78.82±0.76** | **73.86±1.34** | **81.19±1.13** | **79.82±0.90** | **93.49±0.36** | **87.51±1.97** | **94.82±1.76** | **92.05±1.64** |

**Table 3: Ablation study of FedST in terms of local model's personalized accuracy. The experiment is under synthetic shifts with $1 \times$ clients, equals to # of entities.**

| Methods/Dataset | HAR | HHAR | WISDM | Sleep-EDF | Epilepsy |
|---|---|---|---|---|---|
| FedAvg | 78.02±2.48 | 60.87±2.68 | 62.69±2.37 | 73.87±1.04 | 82.99±1.23 |
| Ours w/o $\mathcal{L}_{opl}$ | 80.47±2.01 | 63.52±0.84 | 63.10±0.16 | 76.38±0.79 | 86.24±1.15 |
| Ours w/o $\mathcal{L}_{vol}$ & $\mathcal{L}_{pol}$ | 88.64±1.92 | 72.26±0.78 | 74.37±1.60 | 87.93±0.21 | 92.64±1.36 |
| Ours w/o $\mathcal{L}_{dol}$ | 85.76±1.31 | 71.30±0.52 | 72.45±0.96 | 85.21±1.10 | 93.52±1.54 |
| Ours w/o $\mathcal{L}_{poa}$ | 90.38±0.75 | 73.56±0.40 | 75.59±1.66 | 88.27±1.07 | 94.12±0.86 |
| Ours w/o frequency view | 91.27±0.39 | 74.35±0.77 | 76.48±1.57 | 87.32±1.20 | 93.55±0.94 |
| Ours FedST | **92.98±1.05** | **75.05±0.36** | **79.98±2.06** | **90.22±0.45** | **95.06±2.11** |

number of entities is shown in Table 1 as # subjects. Each client holds a domain, which means the heterogeneity among each sensor monitoring. We also consider a more distributed scenario as an expanded client number $K$.

- **Temporal feature heterogeneity**: We consider two conditions in the real-world application: natural and synthetic.
  - **Natural shifts**: This shift indicates that the time series is non-stationary, and we follow the original train-test partition without modification.
  - **Synthetic shifts**: In synthetic temporal feature shifts, we formulated a more challenging condition, followed by time series domain adaption & generalization context [3, 15, 46]. We mix the test set across clients. Each client holds a test set from another domain.

**Implementation Details.** For different dataset, we set the number of clients $K$ based on # of subjects (entities) in Table1, along with doubled. However, in Epilepsy, the default entity size is 500, which is not realistic for a common federated learning scenario. So we randomly extract 100 and 200 entities for experiments. Each experiment is run for 100 communication rounds. Because our main contribution is not encoder and backbone model design, for fairness, we follow the existing work [15, 50, 69], using the CNN network to train time series data for five epochs in each round. We set the batch size to 128. We use the SGD optimizer with a momentum of 0.5 and a learning rate of 0.01.

## 5.2 Main Results

**Results on spatial heterogeneity and natural temporal heterogeneity.** We follow the previous works, using an averaged

personalized accuracy to evaluate the performance. Table 2 shows the performance of each baseline. We find that FedST achieves the best results compared to the federated learning approaches in all benchmarks. Concretely, FedST outperforms at least 2.37% accuracy improvement on the Epilepsy dataset, and obtains 14.53% accuracy improvement on Sleep-EDF, which is the largest dataset. It indicates the superiority of our model in dealing with large-scale datasets. Regarding baseline performance, we find that general pFL methods that keep personalized models locally, such as FedProx and FedProto, will have a significant impact on their performance. Prediction head decoupling methods like FedRoD and FedTHE, can usually achieve better results than FedAvg, but the improvement is limited. Flame2Graph, tailored for time series data, but does not consider spatial-temporal feature skew, our method still obtains the advantage.

**Results on spatial heterogeneity and synthetic temporal heterogeneity.** To further evaluate the robustness under feature distribution shifts in the test time, we craft the synthetic temporal heterogeneity of feature distributions in Table 4. Facing the unseen domain feature at the inference face, in terms of clients' average accuracy, FedST has much better FL adaptability than state-of-the-art pFL methods in a test-time adaption federated learning setting. Compared with FL methods equipped with test-time adaption, FedTHE and FedICON, FedST improves prediction accuracy by 12.05% and 11.33% on HAR with multiple clients. We can find that our method still performs well in various baselines, as the performance of other baselines has dropped significantly due to unknown feature drifts during testing.

## 5.3 Ablation Study

To validate the effectiveness of each training strategy in FedST, we compare it with several ablations. Mainly, our ablations include: removing whole orthogonal training module $\mathcal{L}_{opl}$, removing view-wise orthogonal training $\mathcal{L}_{vol}$ and $\mathcal{L}_{pol}$, removing domain-wise orthogonal training $\mathcal{L}_{dol}$, removing test-time adaption $\mathcal{L}_{poa}$ and only considering one view selecting from time and frequency. The results shown in Table 3 turn out that each part of our approach plays an important role and they work together to achieve better performance. Notably, we notice that orthogonal training has the

**Table 4: Results of synthetic temporal feature skew in terms of personalization accuracy (%) of local models on five datasets under different numbers of clients ($K$) (domains). Green/bold fonts highlight the best baseline/our method.**

| Dataset | HAR | | HHAR | | WISDM | | Sleep-EDF | | Epilepsy | |
|---|---|---|---|---|---|---|---|---|---|---|
| Number of Clients ($K$) | 30 | 60 | 9 | 18 | 30 | 60 | 20 | 40 | 100 | 200 |
| FedAvg[2017] | 78.02±2.48 | 74.07±1.16 | 60.87±2.68 | 58.60±1.79 | 62.69±2.37 | 57.09±3.24 | 70.87±1.04 | 63.49±2.72 | 82.99±1.23 | 68.44±3.13 |
| FedProx[2020] | 82.92±0.42 | 80.44±0.77 | 59.18±1.45 | 63.97±0.84 | 63.80±2.16 | 60.79±2.30 | 73.69±2.01 | 62.88±0.48 | 85.64±1.43 | 71.55±0.61 |
| FedDyn[2021] | 76.18±1.13 | 71.09±1.57 | 62.96±2.08 | 61.10±2.17 | 57.28±2.78 | 61.10±2.82 | 75.19±1.18 | 61.16±3.05 | 84.87±0.98 | 72.35±2.95 |
| FedProto[2022] | 81.79±0.27 | 77.37±3.25 | 65.88±2.06 | 64.27±2.37 | 64.10±3.11 | 62.11±2.98 | 77.16±0.40 | 59.30±1.45 | 83.31±3.28 | 67.38±1.18 |
| FedRoD[2022] | 82.11±0.01 | 82.55±0.62 | 67.06±0.48 | 66.32±2.68 | 65.90±1.04 | 63.77±2.51 | 72.04±0.21 | 65.02±2.89 | 87.01±0.78 | 70.24±2.01 |
| FedBN[2021] | 85.17±1.75 | 81.58±2.79 | 66.98±1.08 | 63.30±2.91 | 59.20±1.92 | 63.53±3.14 | 76.42±2.78 | 69.95±1.57 | 85.96±1.22 | 69.60±0.33 |
| AlignFed[2022] | 84.88±0.84 | 82.92±0.44 | 68.45±0.26 | 64.29±2.95 | 65.59±3.28 | 69.32±0.64 | 75.82±2.85 | 67.01±2.93 | 89.07±1.30 | 71.17±0.89 |
| FedFA[2023] | 88.08±0.84 | 85.89±2.18 | 70.05±1.82 | 67.30±1.86 | 65.69±2.13 | 69.67±2.03 | 79.00±1.42 | 69.30±1.98 | 90.62±1.69 | 81.83±1.95 |
| FedTHE[2023] | 86.55±0.54 | 79.42±0.42 | 70.33±0.34 | 61.73±1.79 | 65.95±2.79 | 63.84±1.39 | 78.07±0.39 | 68.00±1.31 | 91.33±1.58 | 75.11±2.56 |
| FedICON[2023] | 87.79±0.54 | 80.14±2.24 | 71.14±0.64 | 69.22±1.02 | 70.23±1.29 | 70.18±2.35 | 77.84±1.27 | 72.24±2.50 | 89.17±1.07 | 83.22±1.66 |
| FLAMES2Graph[2023] | 84.49±0.85 | 85.51±1.27 | 72.03±1.02 | 70.41±1.58 | 66.42±0.69 | 70.88±0.94 | 78.31±0.95 | 69.25±1.12 | 88.24±1.64 | 80.12±1.79 |
| **Our FedST** | **92.98±1.05** | **91.47±1.38** | **75.05±0.36** | **72.63±1.20** | **79.98±2.06** | **75.54±1.14** | **90.22±0.45** | **86.16±1.23** | **95.06±2.11** | **91.35±2.30** |

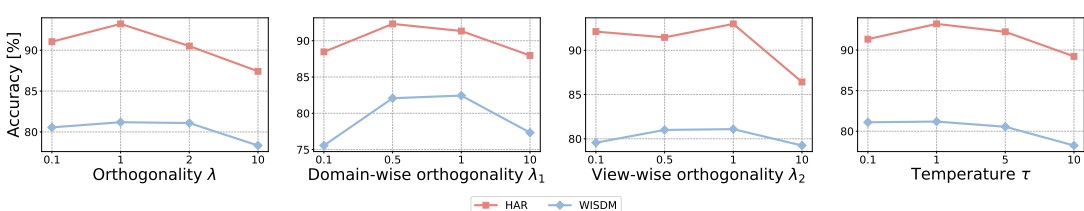

**Figure 5: Performance comparison for hyperparameters under different values in two datasets.**

most significant impact on the improvement. Without orthogonal projections, there is a huge drop in performance of 8.82% to 16.88%. We also find that domain-wise orthogonal constraint and view-wise orthogonal constraint equally contribute to the overall performance, where a minor advantage of domain-wise loss indicates the necessity of decoupling global and local knowledge.

## 5.4 Analysis of Hyperparameters

In our method, the hyperparameter $\lambda, \lambda_1, \lambda_2$ are used to control the strength of the different orthogonal regularization loss. $\lambda$ controls the overall orthogonality constraint, used to balance with the model's predicted cross-entropy loss. As shown in Figure 5, the larger the value of $\lambda$, the more the model focuses on the orthogonality of shared features and personalized features. Through experiments, we find that keeping around 1 for regularization is more appropriate. The $\lambda_1, \lambda_2$ are used to control the orthogonality between views and the orthogonality between clients. We found that the orthogonality of features between clients is more sensitive and plays a dominant role in feature-skewed federated learning. We also studied the contrast temperature of contrastive learning $\tau$. Higher temperature values will increase the output of the model and be more diverse, and the steady-state model also requires a temperature of around 1.

## 5.5 Robustness on Multi-modal Time Series

Our method leverages time and frequency domain information to enhance training. In real applications, many time series data are multi-modal, rather than having only one modality in the original time domain. We extend our method to the multi-modal domain. When we calculate features or losses in the time and frequency domains, we add other modalities in the same way. We compare our

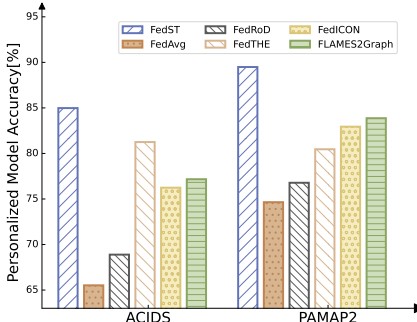

**Figure 6: Accuracy comparison of different baseline on multimodal time series datasets**

method with baselines on two commonly used multi-modal datasets ACIDS and PAMAP2 [49]. From the results shown in Figure 6, we find that it consistently outperforms state-of-the-art baselines with significant advantages, ranging from 3% to 6%, and proved to be valuable in the multi-modal time series domain.

## 6 Conclusion

In this paper, we introduce FedST a federated learning framework for time series that addresses both inter-client and intra-client feature shifts. FedST considers both time and frequency view features. To tackle spatial feature heterogeneity, we apply an orthogonal training paradigm to disentangle the features between views and clients. For temporal feature heterogeneity caused by non-stationarity of time series, we update the model with prototypes in orthogonal subspace to enhance prediction confidence. Experimental results on five datasets demonstrate FedST effectiveness.

# Acknowledgments

This work is supported by the National Key R&D Program of China (Grant No. 2022ZD0160703), the National Natural Science Foundation of China (Grant Nos. 62202422, 62372408 and 62071330), Zhejiang Key Laboratory of Accessible Perception and Intelligent Systems, and Shanghai Artificial Intelligence Laboratory.

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
