# OpenReview forum: "Spatio-temporal Heterogeneous Federated Learning for Time Series Classification with Multi-view Orthogonal Training"
_acmmm.org/ACMMM/2024/Conference — MM2024 Poster_

### Official Review · Reviewer_1gfE · 2024-05-20

**Rating:** 2
**Confidence:** 2

**Summary:**

This paper proposes a FedST framework to handle the spatial and temporal heterogeneity in federated learning for time series. It mainly utilizes an orthogonal feature decoupling scheme across different views and clients. Extensive experiments on five datasets demonstrate its effectiveness. However, this paper only include single modal data (time-serises), for the suitability it should be b) unimodal/unimedia in nature but of sufficient interest to the multimedia community. Because the author try to consider multi-view method to improve they proposed model.

**Strengths:**

S1. The logic of the paper is clear, and the overall presentation is good.

S2. The background and motivation are clearly explained.

S3. The proposed method shows impressive accuracy improvement on the five datasets.

**Limitations:**

W1. Some typos. E.g.

   -Line 103: "(non-IID)" lacks space
   - Line 288: "concepts shits among clients". Shift?
   - Line 291: "labe skew"
   - Line 342: "{(...}"; "denots"
   - Line 399  Need to capitalize at the beginning title
   - Line 403 Eq.3 should be "."
   - Line 492: "a initial global prototypes"
   - Equation (7): "i=1"
   - Line 844: "𝜆, 𝜆1, 𝜆2is"

W2. The experiment section needs a running efficiency report and a comparison against the baselines.

W3. Please list the specific dataset name for Fig.2 and 3

W4. What exactly are the time encoder and frequency encoder? More explanation is needed in the paper.

W5. The concepts of Local-Global Vision, Time-Frequency Vision, and Shared Subspace Features Similarity in Fig.4 are very abstract; it’s completely unclear just from the figure, and more explanation is needed.

W6. According to Sec.4.3, the federated aggregation process in Figure 4 seems to be divided into two parts; one is the aggregation of local prototypes. What does the aggregation process on the right represent? As far as I know, the author should only upload the weights of the classifier for aggregation. Why describe it with a whole model? Have the author tried the effects of uploading all parameters for aggregation?

W7. I think the challenge the paper proposed of Spatial feature heterogeneity may not be a novel challenge. For example, we have the HHAR dataset, where if the paper consider Subjects as domains, we can analogize that a domain is a hospital, and each hospital has up to six types of case (classes) data. Since data cannot be shared, we use federated learning to solve the problem of data non-sharing, which is exactly the goal for which federated learning was proposed. From the description of Challenge 1, this doesn’t seem like a new problem.

W8. For Eq.10, I can't understand the meaning of \phi_{P{s,p}}, P_{s,p}, and W_{k+1}. I didn’t find the explanation of these notations.

W9. For the ablation experiments, it seems that the frequency view contributes very little to the model, could a single view complete the entire model design? You should also add experiments w/o L_{vol} and w/o L_{pol}.

W10. The hyperparameter experiments section should include experiments on how to set the parameter \delta.

W11. You need to add some tables and figures to show the non-IID distribution of the dataset since improving federated algorithms under non-IID conditions is the main highlight of your work.

W12. Actually Sec.5.5 should be the main part of the paper's experiment since the purpose of ACM MM is to focus more on multi-modal work

**Suitability:**

2

---

### Official Review · Reviewer_q8sE · 2024-05-24

**Rating:** 4
**Confidence:** 4

**Summary:**

This paper proposes a federated learning method called FedST to address the challenge of dual spatial-temporal feature skew in time series classification tasks. The key ideas are:
1) During the training stage, collaborating time view and frequency view of time series data to enrich the mutual information and adopting orthogonal projection to disentangle and align the shared and personalized features between views and between clients.
2) During the testing stage, apply prototype-based predictions and model-based predictions to achieve model consistency based on shared features.
The proposed FedST method is evaluated on multiple real-world time series classification datasets and shows consistent performance improvements over state-of-the-art federated learning baselines.

**Strengths:**

1) The paper identifies the unique dual spatial-temporal feature skew challenge tailored for time series in the real-world application of federated learning.
2) The orthogonal feature decoupling and regularization techniques are designed to effectively handle the challenges posed by spatial and temporal feature drifts.
3) The method is comprehensively evaluated on multiple real-world time series datasets, demonstrating its effectiveness.

**Limitations:**

The paper focuses on time series classification, while there could be other important time series tasks (e.g., forecasting, anomaly detection) that also suffer from the spatial-temporal feature skew issue in federated learning.

**Suitability:**

2

---

### Official Review · Reviewer_8nh7 · 2024-05-24

**Rating:** 4
**Confidence:** 2

**Summary:**

Despite the general exploration of feature distribution skew in Federated Learning (FL), applying it to time series data becomes more challenging due to the dual feature heterogeneity exhibited in both spatial and temporal dimensions. Spatial feature heterogeneity refers to the cross-client feature skew caused by differences in data collection methods among different devices and individuals, while temporal feature heterogeneity refers to the non-stationarity of time series data, meaning that the feature distribution of future data may differ from that of past observed data.

To address these challenges, this paper proposes FedST, a two-stage federated learning framework designed to tackle spatial-temporal feature heterogeneity. The framework primarily consists of two modules: orthogonal training and consistency testing. During the training phase, the paper introduces orthogonal decoupling of cross-view and cross-client representations to extract and utilize both shared and private features. In the testing phase, the paper minimizes uncertainty and fine-tunes representations by aligning prototype predictions with model predictions based on orthogonal subspaces. Through this approach, FedST effectively handles feature skew in time series data within federated learning applications.

In summary, the contributions of this paper include:

1.Introducing the unique challenge of spatial-temporal feature skew in time series data for real-world federated learning applications.

2.Proposing the FedST framework, which leverages orthogonal latent space to decouple client-level and view-level features and employs a prototype-based consistency approach for test-time adaptation.

3.Demonstrating through extensive experimental results that FedST outperforms existing state-of-the-art solutions in personalized federated learning, feature-skewed FL, and time series data FL.

**Strengths:**

1. Addressing Spatio-Temporal Feature Heterogeneity Challenges: The paper introduces the FedST framework, which specifically tackles the heterogeneity of spatial and temporal features in time series data in real-world applications. Such feature heterogeneity often leads to performance degradation and slower convergence in federated learning.

2. Personalized Model Training: FedST employs personalized federated learning (pFL) methods to enhance the performance of client-specific models during local testing, allowing them to better adapt to the unique data distribution and requirements of each client.

3. Multi-Perspective Feature Extraction: The FedST framework improves model generalization and robustness by extracting and utilizing shared and private features from cross-perspective and cross-client representations through orthogonal training and consistency testing.

4. Superior Experimental Performance: Extensive experimental validation demonstrates that FedST outperforms existing state-of-the-art solutions in personalized federated learning, feature-shift federated learning, and time series federated learning, proving its effectiveness and advantages.

**Limitations:**

1. Lack of discussion on privacy protection

2. Whether this method will transmit more parameters, leading to a decrease in communication efficiency

**Suitability:**

2

---

### Official Review · Reviewer_1mz7 · 2024-05-26

**Rating:** 3
**Confidence:** 2

**Summary:**

The paper presents FedST, a federated learning framework specifically designed to address the challenges of spatio-temporal feature heterogeneity in time series classification. The authors focus on the dual spatial-temporal feature skew that arises due to the dynamic nature of time series data collected by distributed sensors. The proposed FedST, leverages both time and frequency views of time series data and employs an orthogonal training approach to disentangle and align shared and personalized features across clients and views.

**Strengths:**

1. The posposed FedST introduces a unique method to handle the complexity of time series data in FL by using a dual-view (time and frequency) orthogonal training mechanism, which tackles both spatial and temporal feature heterogeneity, which are critical issues in real-world applications.

2. This paper have conducted extensive experiments on multiple real-world classification datasets and multimodal time series datasets. The consistent outperformance of FedST against state-of-the-art baselines demonstrates the effectiveness of the proposed method.

**Limitations:**

1. The orthogonal training approach, while effective, may introduce additional complexity into the FL process. The paper could benefit from a discussion on the computational overhead and scalability of the proposed method.

2.  FedST's scalability to a large number of clients and high-dimensional time series data is not thoroughly discussed. The overhead of maintaining orthogonality across a massive distributed system may become a bottleneck, potentially limiting the framework's effectiveness in extremely large federated settings.

**Suitability:**

1

---

### Meta-Review · Area_Chair_PTea · 2024-07-05

**Recommendation:** Accept (Poster)
**Confidence:** 4

**Metareview:**

The paper proposes FedST, a two-stage federated learning framework that effectively tackles spatial-temporal feature heterogeneity in time series data. It demonstrates superior experimental performance but lacks discussion on privacy protection and communication efficiency, making it a borderline accept.